# Diseases of Eucalypts in Paraguay and First Report of *Teratosphaeria zuluensis* from South America

**Ximena Silva** [1,*] , **Jolanda Roux** [2] **and Fred O. Asiegbu** [1,*]

1   Department of Forest Sciences, University of Helsinki, Latokartanonkaari 7, FI-00014 Helsinki, Finland
2   Department of Plant and Soil Sciences, Forestry and Agricultural Biotechnology Institute (FABI),
    Faculty of Natural and Agricultural Sciences, University of Pretoria, Pretoria 0028, South Africa;
    jolanda.roux@gmail.com
*   Correspondence: ximena.silvapalacios@helsinki.fi (X.S.); fred.asiegbu@helsinki.fi (F.O.A.)

**Abstract:** *Background and objectives:* The global forest economy is threatened by eucalypt pathogens which are often latent or cryptic species that escape common quarantine and detection methods. Plantation forestry using eucalypts is of considerable importance to Paraguay, but knowledge regarding the pests and diseases affecting these plantations is limited. This study identified fungal diseases present in these plantations. *Materials and Methods:* We surveyed eucalypt plantations in four provinces in Paraguay and collected material from diseased trees for identification of the causal agents. The samples were analyzed using a combination of morphological and molecular methods. *Results:* Diseases encountered included Botryosphaeria stem canker, Calonectria leaf blight, Chrysoporthe stem canker, myrtle/eucalypt rust, Coniella leaf spot, heartwood rot and Teratosphaeria stem canker. Contrary to expectations, the causal agent of Teratosphaeria stem canker was identified as *Teratosphaeria zuluensis* (M.J. Wingf., Crous & T.A. Cout.) M.J. Wingf. & Crous and not *Teratosphaeria gauchensis* (M.-N. Cortinas, Crous & M.J. Wingf.) M.J. Wingf. & Crous, that is commonly documented for the South American region. *Conclusions:* This study updates the knowledge on forest fungal pathogens in Paraguayan eucalypt plantations and is the first report of *T. zuluensis* in Paraguay and in South America.

**Keywords:** Calonectria leaf blight; Chrysoporthe canker; myrtle rust; alien invasive species; Teratosphaeria stem canker

## 1. Introduction

*Eucalyptus* (Myrtaceae) species are native to Australia but form the basis of plantation forestry industries on multiple continents. More than 20 million hectares are planted globally for the production of pulp, energy and timber, augmenting the declining productivity of native forests [1,2]. Over 30% of eucalypt plantations are in South America [1]. The first plantations of eucalypts for industrial timber in Paraguay were established in the 1940s, and currently, there are approximately 100,000 hectares of eucalypts in the country [3]. Eucalypt plantations in the country are considered profitable due to their fast growth and short rotation period [4]. According to the last census, eucalypt plantations represented around 93% of exotic plantations in Paraguay [5]. They comprise mostly *Eucalyptus grandis* Hill ex Maiden and *E. grandis* hybrids such as *E. grandis* × *urophylla* and *E. grandis* × *camaldulensis* [6,7] These are mostly used for timber and secondarily for energy [8].

In recent years, pests and diseases affecting *Eucalyptus* species and their hybrids grown in plantations globally have become increasingly common, particularly in tropical countries in the Southern Hemisphere [9]. Eucalypt plantations consist mostly of single clone monocultures, which renders them more vulnerable to losses caused by pathogen attack [10,11]. Several host-jumps and host

range expansions of native fungal species in countries to which eucalypts have been introduced have been reported. For example, *Austropuccinia psidii* (G. Winter) Beenken and *Chrysoporthe cubensis* (Bruner) Gryzenhout & M.J. Wingf. are both thought to have originated on native Myrtaceae in South America and adapted to *Eucalyptus* species and later spread globally [12–14]. The possibility of host shifts is enhanced by the presence of related native Myrtaceae tree species, such as in South American countries [15], consequently increasing the potential appearance of novel diseases in these areas [16,17]. Inadvertently, the forestry industry has contributed to the evolution and expansion of pathogen ranges by exposing tree species to microorganisms with which they have not co-evolved by moving vegetative material across continents [10]. Additionally, many microbes are latent pathogens that remain undetected by traditional quarantine methods, resulting in the movement of pathogens between continents [18]. Thus, both local adaptations of native fungal pathogens and the introduction of novel pathogens can affect eucalypt plantations globally [10].

Knowledge of eucalypt diseases in Paraguay is limited. The only fungal species formally reported by peer reviewed journals as pathogens of eucalypts in Paraguay are *Austropuccinia psidii* [19], *Pseudocercospora paraguayensis* (Tak.Kobay.) Crous [20] and *Teratosphaeria eucalypti* (Cooke & Massee) Crous [21]. This lack of knowledge regarding the incidence, identity and severity of diseases affecting eucalypt plantations hinders the effective management of diseases and gives rise to economic losses. In the face of globalization, preventing or preparing for novel arrivals and managing local pathogens require constant monitoring, control and research [10]. The objective of this study was to address this knowledge gap and build a foundation on which future actions can be developed. We surveyed and identified eucalypt fungal pathogens in commercial plantations in Paraguay using a combination of morphological and molecular methods.

## 2. Materials and Methods

### 2.1. Site Description and Sample Collection

Eucalypt plantations in the Eastern Region Provinces of Paraguay were visited at intervals between December 2014 and July 2016 to survey for fungal pathogens of eucalypt trees. Site visits took place in December 2014, August to December 2015 and July 2016, during which time plantations of the INFONA (Spanish acronym of the National Forest Institute) in Itapúa (three compartments) and three private companies were visited. The private companies where samples were taken are situated in Caazapá (21 compartments), Alto Paraná (13 compartments) and San Pedro (three compartments) (Figure 1). The history of the compartments varied but all were typically used for planting eucalypts. The average annual temperature of Alto Parana, Caazapa and San Pedro is 23 °C degrees, while in Itapua is 21 °C. The average rainfall values are 2419 mm/year, 1990 mm/year, 1830 mm/year and 1146 mm/year, for Itapua, Alto Parana, Caazapa and San Pedro, respectively. The average altitude are 225 msl., 200 msl., 195 msl. and 84 msl. in Alto Parana, Caazapa, Itapua and San Pedro, respectively. Soils of the sampled areas are of fine to very fine clay and of very low fertility, and thus, are used mostly for forestry purposes.

In each compartment, haphazard transects were walked in order to find diseased trees. The duration of time spent per compartment depended on the time available at the site and the abundance of any diseases observed. Where encountered, leaf and stem samples displaying symptoms of disease were collected from representative trees in each plot. On average, 5–10 symptomatic leaves were collected per affected individual and 1–3 stem samples per symptomatic tree, from 5–10 trees per compartment. Samples were collected from a total of 255 eucalypt trees of varying age classes.

Two forestry nurseries were surveyed for diseases. These were located in Alto Parana and Caazapa. From these nurseries, 65 saplings younger than a year were collected. Samples were placed into individual bags for each tree and transported to a laboratory for further processing.

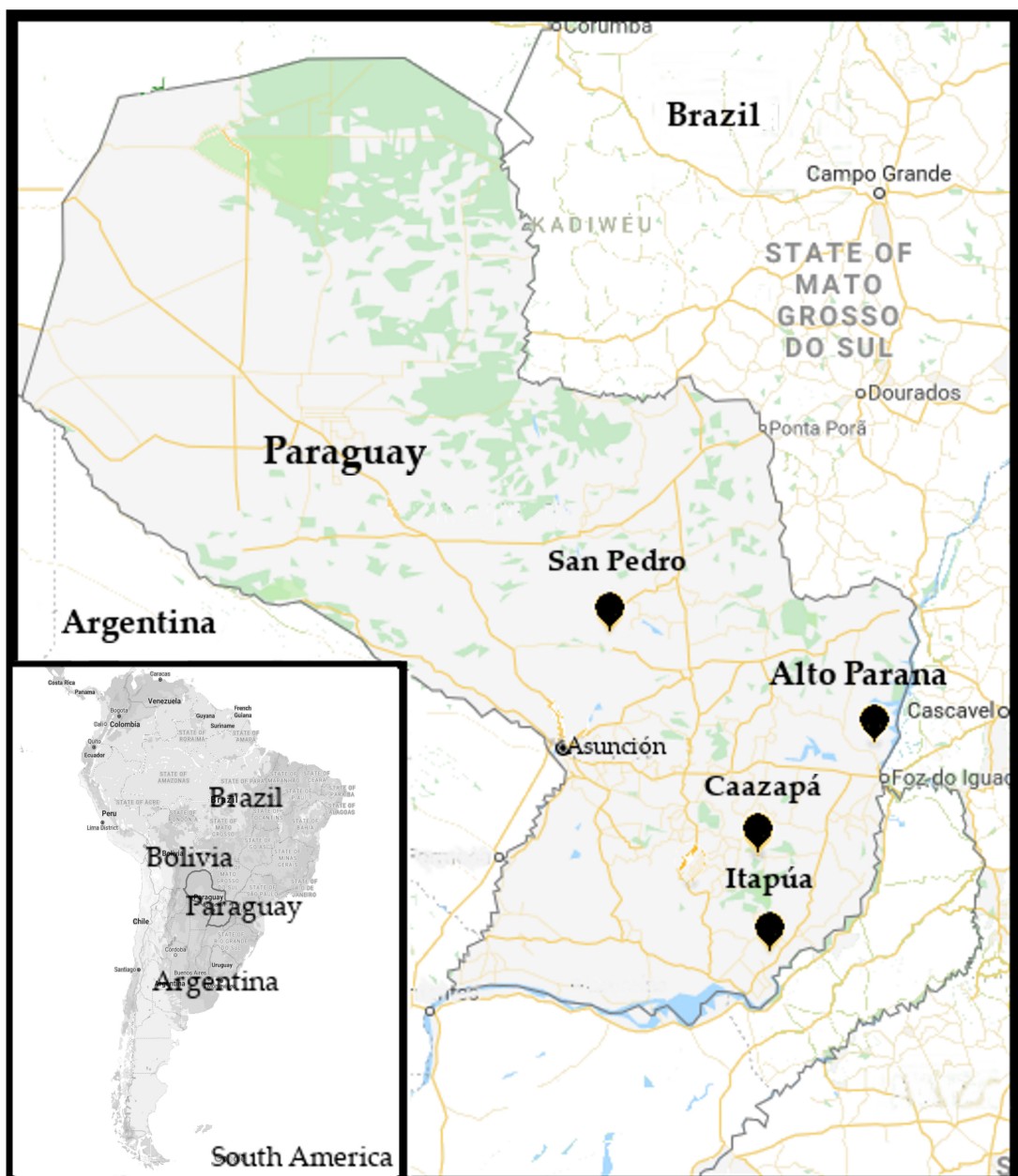

**Figure 1.** Map of Paraguay showing the geographical locations of San Pedro, Alto Parana, Caazapa and Itapua from which eucalypts samples were collected. Map obtained from Google maps (Google, Mountain View, CA, USA).

*2.2. Isolation and Culture Purification*

Two isolation strategies were used to obtain the causal agents of the disease symptoms observed on the collected samples. This included the collection of spores from fungal fruiting bodies on the surfaces of diseased material and the transfer of the leading margins of lesions to a general fungal culture medium. In order to isolate single spores of potential pathogens associated with the disease symptoms, moisture chambers and standard isolation methods were used as previously described (Maússe-Sitoe et al., 2016). Purification of the fungal isolates onto 2% malt extract agar (MEA, Sigma-Aldrich, VWR International LLC Laboratory supplies) was done at the laboratories of the SENAVE (Spanish acronym of the National Service of Plant and Seed Quality and Health) in San Lorenzo. Purified cultures were grouped into morphotypes based on the color of mycelia, spore morphology and growth traits.

Representative isolates from each morpho-group were selected for identification with the aid of DNA sequencing. All isolates were deposited in HAMBI mBRC (Microbial Domain Biological Resource Centre HAMBI), University of Helsinki, Finland, and selected isolates were additionally placed in the CMW fungal culture collection at FABI (Forestry and Agricultural Biotechnology Institute), University of Pretoria, South Africa.

### 2.3. Nucleic Acid Extraction, PCR and Sequencing

DNA extraction and sequencing were conducted at the Department of Forest Sciences, University of Helsinki, Finland for all samples, with the exception of biotrophic pathogens which were processed and analyzed at the University of Pretoria, Pretoria, South Africa. DNA was extracted following a CTAB protocol [22].

Three gene regions were selected for DNA sequencing: the fungal barcode region, the internal transcribed spacer (ITS) region of the rRNA operon, the Beta tubulin (BT2) and the translation elongation factor 1-$\alpha$ gene (tef). The ITS 1 and 2 regions were amplified using the universal primers ITS1-F [23] and ITS4 [24], the Beta tubulin (BT2) region was amplified using BT2A and BT2B [25], and the translation elongation factor 1-$\alpha$ using primers EF1-728F and EFI-986R [26]. The ITS regions were amplified with PCR according to the method described by Cortinas et al. [27], which was also followed for the amplification of the BT2 region of the Teratosphaeria canker samples. The method outlined by Myburg et al. [28] was followed for the amplification of the BT2 region for the Chrysophorte canker, while for the TEF region, the method described by Slippers et al. [29] was followed. Dream Taq polymerase was used according to the manufacturer's recommendations (Thermofischer Scientific, Denmark). Samples were commercially sequenced at Macrogen, Inc. (Amsterdam, Netherlands) through standard sanger sequencing on a ABI 3730xl DNA Analyzer (96-capillary type) and using BigDye® Terminator v3.1 Cycle Sequencing Kits (Applied Biosystems, Foster City, CA, USA).

For samples representing putative rust fungal species, DNA sequence analysis and morphology was used to establish the identity of the rust pathogen. Genomic DNA was extracted from a single uredinium per host using the Ultraclean® Microbial DNA Isolation Kit (MoBio Laboratories, Carlsbad, CA, USA). Rust samples were sequenced at the DNA Sequencing Facility of the Faculty of Natural and Agricultural Science (NAS), University of Pretoria, South Africa.

### 2.4. Sequence Analysis and Isolate Identification

All obtained sequence electropherograms were manually quality checked using Geneious R11 [30]. Thereafter, Blastn searches were conducted in the NCBI database (National Center for Biotechnology Information, U.S. National Library of Medicine, Bethesda, MD, USA) to obtain a first level identification of sequences. Because ITS and BT sequences were sufficient for accurate identification of the Teratosphaeria stem canker and rust samples, sequences of these fungi were further analyzed using phylogenetic analyses. Sequences were aligned using the E-INS-i strategy in MAFFT version 7 [31] and adjusted in MEGAx [32]. Sequences of different regions were concatenated with Mesquite 3.6.1 [33] to obtain a combined tree. The Maximum Likelihood method and Kimura 2-parameter model [34] and Neighbor-Join and BioNJ algorithms in MEGAx were used for tree construction. Phylogenetic trees were viewed using MEGAx. Outgroups and comparative sequences were selected according to published literature [27,35,36].

## 3. Results

### 3.1. Observed Diseases

Diseases observed in the field included both stem and leaf diseases (Table 1). Symptoms of leaf diseases were typical of myrtle/eucalypt rust, Calonectria leaf blotch, Coniella leaf spot and Mycosphaerella leaf blotch. Myrtle/eucalypt rust symptoms included leaf spot, shoot and leaf death. Yellow masses (Figure 2a) of uredioniospores were commonly encountered on affected leaf and shoot

parts. This disease was encountered on *Eucalyptus grandis*, *E. urophylla* × *grandis*, and *E. benthamii* Maiden & Cambage in Caazapa (Table 1). Leaf spots with pycnidia distributed in concentric rings, typical of Coniella leaf spot (Figure 2b), were observed on *E. grandis* and *E. benthamii* in Caazapa; however, the causal agent could not be isolated. Leaf blotch lesions coalescing on mature leaves, consistent with Calonectria leaf blotch symptoms (Figure 2c), were found on *E. benthamii* in Caazapa (Table 1). Leaf spots delimited by leaf veins and pycnidia on the underside of leaves, consistent with Mycosphaerella leaf spot, were found on *Eucalyptus urophylla* × *grandis* in nurseries in Alto Parana and Caazapa, on *E. urophylla* × *camaldulensis* and *E. grandis* in nurseries at Caazapa and on adult leaves of *E. grandis*, *E. urophylla* × *grandis* and *E. grandis* × *camaldulensis* in San Pedro (Table 1).

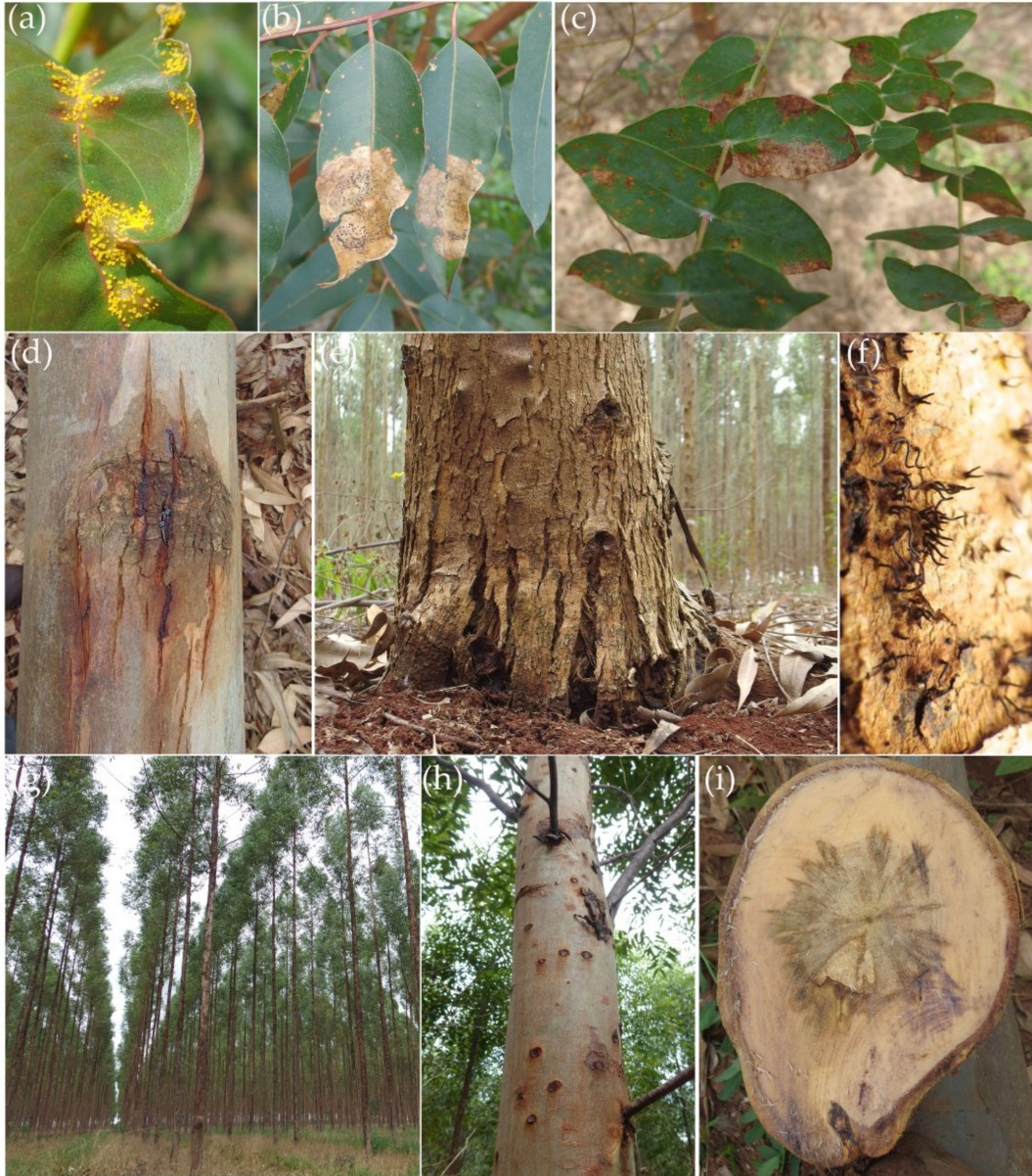

**Figure 2.** Symptoms of fungal diseases in Paraguayan eucalypts plantations. (**a**) Myrtle/Eucalypt rust symptoms on mature leaf, (**b**) Coniella leaf spot, (**c**) Calonectria blight, (**d**) Botryosphaeria canker, (**e**) Chrysoporthe canker, (**f**) *Chrysoporthe* sp. on eucalypt bark, (**g**, **h**) Teratosphaeria stem canker, (**i**) cross section showing a stem disease caused by unknown pathogen present in *Eucalyptus urophylla x grandis*.

**Table 1.** Sampled eucalypts plantation areas in Paraguay, host species, affected plant parts and fungal pathogens isolated from them.

| Provinces | Towns | Coordinates | Host Species | Affected Plant Parts | Pathogens |
|---|---|---|---|---|---|
| Alto Parana | Hernandarias | 25°22′00″ S 54°45′00″ W | *Eucalyptus urophylla × grandis* | Sapling's leaves | *Pallidocercospora* sp., *Pseudoteratosphaeria* sp. |
| | | | | Stem | *Botryosphaeria* sp., *Chrysoporthe* sp., *Teratosphaeria zuluensis* |
| | | | *Eucalyptus grandis* | Sapling's shoot | *Botryosphaeria* sp. |
| | | | | Stem | *Chrysoporthe* sp., *Teratosphaeria zuluensis* |
| Caazapá | San Juan Nepomuceno | 26°07′00″ S 55°55′48″ W | *Eucalyptus urophylla × grandis* | Sapling's leaves | *Pseudoteratosphaeria* sp. |
| | | | | Leaves | *Austropuccinia psidii* |
| | | | | Sapling's shoot | *Botryosphaeria* sp. |
| | | | | Stem | *Botryosphaeria* sp., *Teratosphaeria zuluensis* |
| | | | *Eucalyptus urophylla × camaldulensis* | Sapling's leaves | *Pseudoteratosphaeria* sp. |
| | | | *Eucalyptus camaldulensis* | Stem | *Botryosphaeria* sp., *Chrysoporthe* sp. |
| | | | *Eucalyptus grandis* | Leaves | *Austropuccinia psidii* |
| | | | | Stem | *Botryosphaeria* sp., *Chrysoporthe* sp., *Teratosphaeria zuluensis* |
| | | | *Eucalyptus grandis × camaldulensis* | Stem | *Teratosphaeria zuluensis* |
| | | | *Eucalyptus benthamii* | Leaves | *Austropuccinia psidii*, *Calonectria* sp. |
| | | | *Corymbia citriodora* | Stem | *Teratosphaeria zuluensis* |
| Itapua | Pirapo | 26°51′36″ S 55°32′42″ W | *Eucalyptus saligna* Smith. | Leaves | *Botryosphaeria* sp. |
| | | | | Stem | *Botryosphaeria* sp. |
| | | | *Eucalyptus urophylla × grandis* | Leaves | *Botryosphaeria* sp. |
| | | | | Stem | *Botryosphaeria* sp. |
| San Pedro | Vlla del Rosario | 24°25′28.3″ S 57°06′20.8″ W | *Eucalyptus grandis* | Leaves | *Pseudoteratosphaeria* sp. |
| | | | | Stem | *Teratosphaeria zuluensis* |
| | | | *Eucalyptus urophylla × grandis* | Leaves | *Pseudoteratosphaeria* sp. |
| | | | | Stem | *Teratosphaeria zuluensis* |
| | | | *Eucalyptus grandis × camaldulensis* | Leaves | *Pseudoteratosphaeria* sp. |

Stem diseases encountered were Botryosphaeria stem canker, Chrysoporthe canker, Teratosphaeria stem canker and an unidentified wood rot disease. Typical symptoms of Botryosphaeria stem canker included longitudinal cracks in the bark and kino exudation and was observed in Alto Parana, Caazapa and Itapua (Table 1, Figure 2d). Chrysoporthe canker symptoms such as swelling, cracking and splitting of the bark at the bases of the trees, as well as superficial pycnidia (Figure 2e–f) and yellow spore masses, were found in Caazapa and Alto Parana. Symptoms typical of Teratosphaeria stem canker disease, namely discrete, sunken, oval necrotic lesions, kino exudation (Figure 2g–h) and dark pycnidia were encountered at Alto Parana, Caazapa, and San Pedro (Table 1). This disease resulted in tree stems having a red to black color as a result of the kino exudation from lesions. A rot disease affecting the heartwood of mature six year old *E. urophylla × grandis* trees was observed at Caazapa. The symptoms were cracks in the bark, and when cut, streaking patterns of discoloration were observed within the heartwood (Figure 2i). The causal agent for this disease could not be isolated.

### 3.2. Pathogen Identities

Fungi obtained from the sampled material were morphologically typical of the fungal genus responsible for the observed field symptoms. Cultures from the spores isolated from Calonectria leaf blotch samples resulted in fast growing orange to brown colonies with elongated and ellipsoidal conidia typical of Calonectria species. The spores isolated from the Mycosphaerella leaf spot samples were elongated Cercospora-like and hyaline and slightly curved, and resulted in very slow growing, olivaceous grey cultures. Spores from the Botryosphaeria canker samples were hyaline and ellipsoid to fusiform, and colonies were fast growing and turned grey to black with age. The observed perithecia on the Chrysoporthe canker samples were dark, while the spores were hyaline and fusoid to oval, resulting cultures that were fast growing and white to orange and fluffy. The spores isolated from the Teratosphaeria canker samples were hyaline and oval, and resulted in cultures that were very slow growing and olivaceous green.

Blastn results confirmed the identities of the fungi associated with the eucalypt diseases in Paraguay as made based on field symptoms. Leaf pathogens, based on Blastn comparisons, could be assigned to the genera *Austropuccinia*, *Calonectria*, *Pallidocercospora* and *Pseudoteratosphaeria* (Table 2). Based on phylogenetic analyses of the ITS gene region of rust samples from Paraguay, the cause of myrtle rust in the country could be confirmed as being *A. psidii* (Supplementary Materials, Figure S1).

**Table 2.** Fungal pathogens in Eucalypt plantations and nurseries in Paraguay with their Genbank accession numbers and herbarium data.

| Isolate | GenBank Accession No. of Isolate | GenBank Accession No. of Best Matches | Mi (%)/Qc (%) | Suggested Name for Isolate | Region | Herbarium |
|---|---|---|---|---|---|---|
| PUCPY01 | MH513947 | KF792096 | 99/100 | *Austropuccinia psidii* | ITS | |
| BOTPY01 | MH518155 | MN856397 | 99/99 | *Botryosphaeria* sp. | ITS | HAMBI |
| | MN508824 | KR232664 | 100/100 | *Botryosphaeria* sp. | TEF | |
| CALPY01 | MH512924 | MH863819 | 98.85/100 | *Calonectria* sp. | ITS | HAMBI |
| CRYPY01 | MH518146 | AF273476 | 99.79/100 | *Chrysoporthe* sp. | ITS | HAMBI |
| | MN508822 | KX639104 | 100/100 | *Chrysoporthe* sp. | BT2 | |
| MYCPY01 | MH518178 | GQ852745 | 98.34/100 | *Pallidocercospora* sp. | ITS | HAMBI |
| TEPY01 | MH518183 | EF394846 | 98.75/99 | *Pseudoteratosphaeria* sp. | ITS | HAMBI |
| CMW49670 | MN508828 | JQ732915 | 100/100 | *Teratosphaeria zuluensis* | ITS | CMW fungal culture collection at FABI, HAMBI |
| | KY401245 | DQ240110 | 100/100 | *Teratosphaeria zuluensis* | BT2 | |

Comparison of fungal isolate with best matches in GenBank. ITS, internal transcribed spacer, BT2, beta tubulin; TEF, elongation factor; Mi, maximum identity; Qc, query coverage. All samples were deposited at HAMBI mBRC (Microbial Domain Biological Resource Centre HAMBI), University of Helsinki, Finland, and the *Teratosphaeria zuluensis* samples were also deposited at the CMW fungal culture collection of FABI (Forestry and Agricultural Biotechnology Institute), University of Pretoria, South Africa.

Stem canker pathogens were assigned to the genera *Botryosphaeria*, *Chrysoporthe* and *Teratosphaeria* (Table 2). Blastn and phy logenetic analyses of the Teratosphaeria stem canker isolates using ITS and BT2 showed that isolates from Paraguay represented *T. zuluensis* (M.J. Wingf., Crous & T.A. Cout.) M.J. Wingf. & Crous (Table 2, Figure 3).

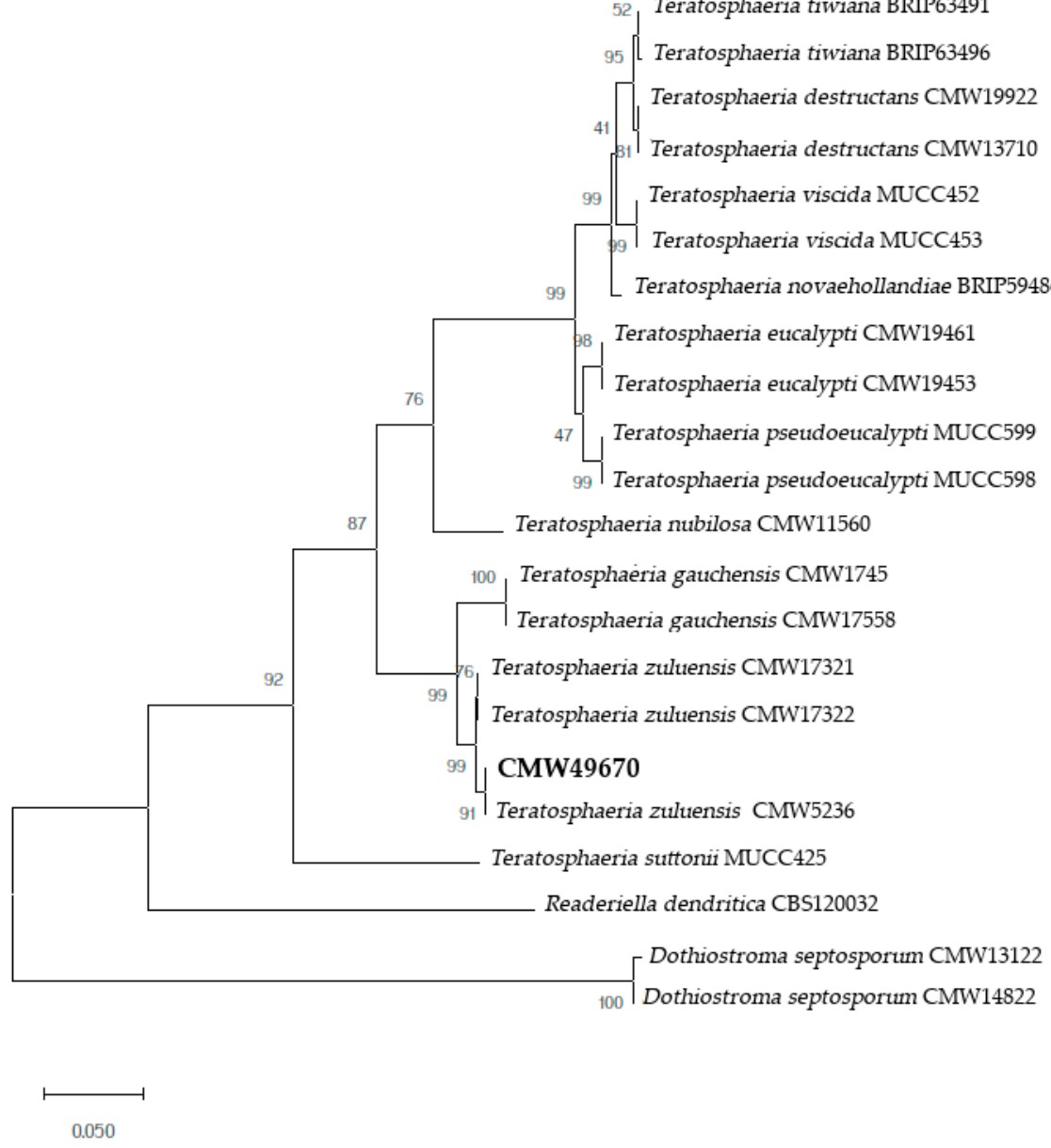

**Figure 3.** Maximum likelihood tree of Teratosphaeria canker isolates from Paraguay based on the combined analysis of ITS and the BT2 regions. Isolates and their culture collection names are indicated as CBS (Dutch Centraalbureau voor Schimmelcultures), CMW Fungal Culture Collection at FABI, MUCC (Murdoch University culture collection) and BRIP (Biosecurity Queensland Plant Pathology Herbarium), respectively. Bootstrap confidence values (1000 replicates) are indicated above the internodes. The tree was rooted in *Dothiostroma septosporum.*

The leave diseases found in the nurseries were associated with *Pallidocercospora* sp. and *Pseudoteratosphaeria* sp., while that from the shoots was associated with *Botryosphaeria* sp. (Table 1).

## 4. Discussion

This study provides results of the most detailed survey of the diseases affecting plantation grown eucalypt trees in Paraguay to date. Diseases encountered included Botryosphaeria, Chrysoporthe and Teratosphaeria stem canker, and an unidentified stem rot disease, Calonectria leaf blotch, Coniella leaf spot, Mycosphaerella leaf spot and myrtle/eucalypt rust. With one exception, none of the diseases were new to South America, yet reports from Paraguay extend their geographic range in the region. Of significant importance was the discovery of *Teratosphaeria zuluensis* as the cause of Teratosphaeria stem canker disease in Paraguay; this represents a new species record for South America.

The disease documented as the most concerning by forestry companies was Teratosphaeria stem canker, as it is reported to negatively affect both wood quality and yield and increase pulping costs. Teratosphaeria stem canker is most commonly referred to as 'ox foot' by foresters in Paraguay. The disease was recorded in all provinces surveyed, with the exception of Itapua. Because of the reported importance of the disease, the ITS and BT2 regions of the causal agent were sequenced, resulting in the surprising discovery that the disease in Paraguay is caused by *T. zuluensis* and not *T. gauchensis* (M.-N. Cortinas, Crous & M.J. Wingf.) M.J. Wingf. & Crous as recorded in neighboring countries. Previously, the distribution of *T. zuluensis* was limited to China, Malawi, Mozambique, Mexico, South Africa, Thailand, Uganda, Vietnam and Zambia [37].

The discovery of *T. zuluensis* in Paraguay is a novel finding of importance for the forest industry and forest health authorities. An implication of this finding is that both Teratosphaeria stem canker pathogens are present on the South American continent. The relevance of the proximity of these two related pathogens is that they have the potential to hybridize and give rise to a new lineage or transfer genes, both of which could result in more virulent pathogens and subsequent economic losses [38,39]. Presently, there is no clear indication of the point of entrance of this pathogen. Species in the Teratosphaeriaceae have been shown to have been moved globally with the expansion of eucalypt plantation forestry. It was, for example, shown that *T. zuluensis* occurs in the seed of *E. grandis* trees [40]. Species in the Teratosphaeriaceae have also been identified as symptomless endophytes in the leaves, twigs and stems of *E. grandis* trees [37,41]. Further studies on the population diversity of Paraguayan *T. zuluensis* isolates may shed light on its variability and potential origin(s).

*Austropuccinia psidii* has previously been recorded in Paraguay from *Syzygium jambos* (L.) Alston, *Psidium guajava* L. [19,42] and *Eucalyptus* sp. [43]. The current study is the first to report *A. psidii* from *E. benthamii* in plantations. Previous studies in Brazil have reported that *E. benthamii* is susceptible to *A. psidii*, but did not report its occurrence in the field [44,45], making this the first study to do so.

Furthermore, more detailed phylogenetic studies are required to confirm the species identities of the other pathogens encountered in this study. Although multiple gene regions were sequenced for selected isolates, none were sufficient to satisfactorily identify, e.g., species in the Botryosphaeriaceae or *Calonectria* at a species level. This is the first study to report *Pseudoteratosphaeria* and *Pallidocercospora* from eucalypts in Paraguay, although species in these genera have previously been observed on eucalypts in other countries in South America [46,47]. Similarly, stem canker in eucalypts related to species belonging to the genera *Botryosphaeria* [48] and *Chrysoporthe* [49] recorded in this study are common to South American eucalypt plantations. Further studies are necessary to confirm the identity of these species and determine how significant each of these pathogens may be for the local forestry industry. This is of particular relevance for *Chrysoporthe*, because *C. cubensis* remains recorded as absent in the list of quarantined pathogens for the COSAVE (Spanish acronym of the Southern Cone Plant Health Committee) [50].

The aim of this study was to provide baseline information regarding the incidence and identity of fungal diseases affecting eucalypt plantations in Paraguay in order to better inform disease management strategies. Although this has been achieved, with the recording of *T. zuluensis* as the causal agent of Teratosphaeria stem canker in Paraguay, the report of *A. psidii* on *E. benthamii* in the field and the report of species of *Pallidocercospora, Pseudoteratosphaeria* and *Chrysoporthe* in Paraguayan eucalypts, much remains to be done. The causal agents of the Coniella-like leaf spot disease and the heartwood

disease merit further study. Future studies should also focus on the seasonality of disease occurrence. Although in this study sampling was done during either summer-winter or winter-summer periods, more detailed documentation of the disease incidences and patterns at different periods of the year is required.

The diseases reported in this study may negatively impact forestry production in Paraguay, and thus, forest disease management is of the utmost importance. A first step in forest pathogen management is quarantine, which is the focus for SENAVE and COSAVE, which produce lists of quarantine plant pests for the country and region respectively, and have conventions and treaties to prevent the arrival of various pests and to detect them if present [50,51]. Eradication can be attempted if a pathogen is detected early in a region or country, but this is hindered by the considerable size of forest ecosystems and the lack of human resources to constantly monitor them [52]. To alleviate this lack of personnel and improve local scientific literacy and capacity, citizen science projects can be a useful management tool for pest monitoring to detect the presence and expansion of a pathogen [53] and improve models of pathogen behavior [54] and detection of host recovery [55]. Another pest management tool is avoiding the spread of a pathogen to other areas inside a country by practices such as burning, as seen in the case of eucalypt seedlings infected with *Leptocybe invasa* in Paraguay [56]. Chemical control or biocontrol are common options for nursery material, but for mature trees, silvicultural management and the selection of resistant provenances, families or clones is more effective [57]. Tree species or genotypes need replacement if proven to be highly susceptible to a new pathogen. This necessitates that eucalypt breeding programs diversify and ensure pipelines of genotypes that are genetically heterogeneous [58,59]. The resistant genotypes can replace others more susceptible to diseases or pests. Conventional breeding programs can improve efficiency by the use of modern technologies such as biomarkers, genomic selection, genetic modification and "omics" techniques. These molecular tools could help traditional breeding programs in the selection and design of resistant genotypes, resistant traits, understanding of tree defense mechanisms, host-microbe interactions and the importance of tree microbiomes in tree defense [60–63].

To address the complexities of forest pathogens, Paraguayan institutions should invest in building local capacity and forming forest pest research centers to engage with regional and global pest management initiatives where countries openly share their expertise and knowledge regarding forest pests and pathogens [11,64]. Additionally, collaboration is needed among experts of different disciplines such as geneticists, tree breeders, foresters, landscape ecologists and conservation biologists to fill the interdisciplinary gaps in tree pathogen research [65]. These centers and multidisciplinary teams would better enforce phytosanitary regulations and quarantine methods, which have been shown to effectively reduce the introduction of new pathogens and/or virulent genotypes [66]. Reports of undocumented eucalypt seedlings of unknown origin entering Paraguay are not rare [7,56], and could be a source of new pathogen species or genotypes. Evidence indicates that insufficient attention to quarantine and repeated introductions of a pathogen contributes to its diversification, epidemic outbreaks and economic losses in eucalypt plantations [10]. Breeding trees for resistance is a long and costly process, and it is essential to limit the genetic diversity of forest pathogens and thus prolong the lifespan of tree genotypes [67]. A diverse forest plantation can be more resilient to pathogen invasion and reduce the economic loss caused by pathogens [68], but persistent introduction and diversification of pests and pathogen populations can be harder and more costly to manage [69]. This is particularly relevant because in this study we identified a novel pathogen for this area, *T. zuluensis*; the potential reintroduction and diversification of this pathogen, or the potential hybridization with related *T. gauchensis*, present in neighboring countries, could cause considerable losses to eucalypt plantations, and thus should be avoided.

## 5. Conclusions

This survey lays an important foundation on which forestry companies, the Paraguayan forest department and quarantine institutions can improve and build strategies for the protection of forest

resources in the country. Although most of the diseases encountered in Paraguay are not new to South America, potential novel species may be present in Paraguay, as illustrated by the discovery of *T. zuluensis* in this study. It is, therefore, important to further identify the pathogens obtained in this study at the species level. Further analyses of these pathogens will increase the understanding of the geographical distribution of South American eucalypt pathogens. This is relevant because eucalypt pathogens keep expanding regionally and globally [10]. While phytosanitary regulations can help deter the illegal movement of vegetative material and the subsequent dispersion and arrival of pathogens and new virulent genotypes, common regional and global efforts are needed to strengthen and enforce these regulations.

**Supplementary Materials:** The following are available online at http://www.mdpi.com/1999-4907/11/10/1035/s1, Figure S1: Phylogram based on the analysis of ITS region for *Austropuccinia* rust found in *eucalypts* leaves in Paraguay. Bootstrap confidence values (1000 replicates) are indicated above the internodes. The tree is rooted in *Dasyspora segregaria*

**Author Contributions:** X.S., J.R. and F.O.A. conceived the study. X.S. and J.R. conducted the field surveys and collections. X.S. did the lab work. X.S., J.R. and F.O.A. contributed to the data analysis and interpretation. X.S. drafted the manuscript. J.R. and F.O.A. reviewed and edited the manuscript. F.O.A. acquired funding and provided resources. All authors have read and agreed to the published version of the manuscript.

**Funding:** This research was funded by Academy of Finland (Grant no 307580), the University of Helsinki Chancellor grant, and the University of Helsinki graduate school in Plant Sciences (DPPS). The Programa Nacional de Becas de Postgrado en el Exterior "Don Carlos Antonio López" (BECAL) provided the scholarship to XS. Open access funding provided by University of Helsinki.

**Acknowledgments:** We acknowledge Ginna Granados of University of Pretoria for DNA isolation of the rust samples and laboratory assistance during a research visit by XS to the University of Pretoria. The staff of INFONA, Guyra Paraguay, SENAVE, POMERA and Felber Forestal are acknowledged for their support in this project, in particular Raquel Acuña, Alberto Yanosky, Cecilia Pizzurno, Alfredo Gricyuk, Adriana Caballero, Carolina Garay, Ricardo Kiriliuk, Mauricio Moran, Diana Ramirez, Carlos Arevalos and Jorge Paiva.

**Conflicts of Interest:** The authors declare no conflict of interest.

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
