# Peer review of "Diseases of Eucalypts in Paraguay and First Report of Teratosphaeria zuluensis from South America"

_forests, doi:10.3390/f11101035_

Round 1

Reviewer 1 Report

Summary: This study reports the results of a survey conducted in Eucalyptus plantations from four geographic regions in Paraguay to provide information on fungal diseases present in these forests. The authors report that this information was not previously available, and its acquisition is relevant for the management of forest diseases to support the forest industry in Paraguay.  The study uses standard methods of fungal isolation, and morphological and molecular identification of diseases on plant material and isolates grown out of collected material. The authors discuss the importance of pathogen surveys and the implications for managing forest diseases.

The novelty of the study:

  • This study provides first records of the presence of fungi that are implicated in Eucalyptus diseases (first study to report Pseudoteratosphaeria and Pallidocercospora in Paraguay) and a new host-pathogen association record for Paraguay ( psidii from E. benthamii).
  • Importantly the authors discover a novel Teratosphaeria species (Teratosphaeria zuluensis) causing disease in Paraguay, whereas the disease is typically caused by a different species in other countries. According to the authors, this is the first report of this fungal pathogen in Paraguay and South America.
  • The authors also report the presence of disease (a stem rot) where associated fungi could not be isolated therefore identified but are potentially impacting the growth of trees in one area. This is an interesting finding and some mention of future investigation should be added.

Possible improvements:

  • Discussion of seasonality of collections and disease expression might be useful. From the collection dates provided in the methods it appears that these were performed in the changing seasonality periods between summer-winter or winter-summer, but no descriptions were provided. As well, nothing is mentioned about the seasons that these diseases detected were commonly observed in other locations and how these compare. The environment (in which season and climate are one part of) is important aspect of the disease triangle.
  • The abstract states that fungal isolates were identified using morphological and molecular techniques. Morphological descriptions of diseased material are described in section 3.1, including the spores from fungi on the diseased material. However, no morphological results are reported for the isolates grown out of the diseased material; only that they are identified using molecular techniques. This should be clarified.
  • Isolation of fungi was performed on only one type of media and this may have limited the isolation of other fungi contributing to disease also, especially in the cases where fungi could not be isolated. Were other media trialled? If so this should be mentioned, if not then maybe discuss why. Oomycetes can also be associated with Eucalyptus e.g. Phytophthora captiosa, Phytophthora fallax (Dick et al. 2006) and Nothophytophthora species too (Jung et al. 2017, Studholme et al. 2019). This could be discussed.
  • Page 3, line 103 – what kind of DNA sequencing was performed/what was the model of sequencer?
  • Page 6, line 159 – Blastn confirmed the identities of the fungi found associated with disease, but not the diseases. This needs clarifying.
  • Tables, general comment – the format of these tables needs tidying, the font is different, and they are too big for the pages.
  • Table 1 – the samples that did not yield isolates should ideally be listed too.
  • Table 2 – column titled “suggested name for isolate” should say “suggested identification of isolate”.
  • Figure 1 legend reports Diaporthe septosporum as the out-group but the tree has Dothistroma septosporum. The error needs correcting.
  • I think that SUPP Fig 1 could actually be embedded in the main document as it does add value. I was looking for the regions for sampling and think other readers would expect to see it in the main document. Also unsure why the Supplementary fig’s are within the main manuscript and not a separate document?

Overall:

This is an important piece of work for the forest pathology of Paraguayan eucalyptus forests and will be useful for future diagnostics and management of disease. I enjoyed this manuscript and recommend that it is published after the minor revisions, suggested above, are addressed.

Reviewer 2 Report

The manuscript " Diseases of Eucalypts in Paraguay and first report of
Teratosphaeria zuluensis from South America
"is very interesting regarding the methods used to evaluate the presence of the pathogens from Eucalyptus plantations from Paraguay.
The handling of DNA analyses appeared cautions and thought through. The manuscript uses good English, is easy to read, the style is concise and the information is clear.
The introduction is well written, offering enough background for understanding the Eucalyptus challenges referring to pathogens in South America and specifically Paraguay. Despite that, the goal of the paper should be more specific in compliance with the findings. “build a foundation on which future actions can be developed” seems an objective to large and very generalist one despite all very specific approach that has been made in this study.
The section Materials and methods could be improved a lot as referring to sampling and site descriptions. The samples were taken from 37 plots accounting for 4 localities but no information regarding size, age of the trees, canopy cover, tree dimension, type of soils, average temperature, and precipitations are not presented. Also, management neither (rotation: second, third…).
Even if haphazard transects were used for sampling, is important to set a minimum of criteria. The initial impression was that the plots and stands are one and the same thing. Is it? So, it will be very important to provide details about the size of the plots, what surface was covered by transects, and how many trees were investigated.
As a forestry journal, I would like to see more details regarding trees, stands, and their management related to their susceptibility to different diseases and pathogens.
At the end of this section please provide the total number of trees sampled (stem samples) and leaves collected. Also, for some plots there were taken shoots for analyses, why did was not applicable for all plots?
As regarding results, the only novelty of this study is the revealing of one new pathogen for Eucalyptus plantation in Paraguay- Teratosphaeria zuluensis. All other pathogens were already known to affect these tree species in that area. This is why, if authors own information regarding quantitative details about these pathogens, I advise to include them in this section. It also worthwhile to reflect on the following questions:
Why the authors sampled so many plots and not offer information about the proportion of infected/affected trees? Asymptomatic ones were taken into count?
Discussions appear justified, but some inconsistencies were shown:
Pg.7, line 185-186: “The most commonly encountered disease of eucalypts in Paraguayan plantations was Teratosphaeria stem canker disease” - That is a result delineated from this study. What method authors have used to evaluate this? Nevertheless, this should be moved to the Results section and the method should be explained as well.
Pg.7, line 189: the same with “The disease was encountered in all provinces surveyed, with the exception of Itapua.”
Also, some minor comments and recommendations:
Pg. 1, line 38: needed a point after the first parenthesis.
Pg.4, Figure 1. Please use the same color of letters in figure 1(a) to (h).
Pg.5, Table 1. Some miswriting in the first row of table 1 (“affected parts”).
Please provide a brief description of all acronyms used because not all readers are familiar with national/regional organizations (SENAVE, COSAVE, etc.)
All in all, I congratulate the authors for their tremendous input and I would like to see this manuscript published, but despite the high level of genetic analyses and laboratory handling, I consider that the manuscript has serious issues regarding sampling and interpretation of the results. Also, it will worthwhile to adjust the aim of the study to their specific results, not to a generalist objective like the one mentioned in the text.

Round 2

Reviewer 2 Report

Congratulations to the authors! I am satisfied with the improvements made. My decision is to accept the manuscript for publication.